# Trapping an octahedral Ag$_6$ kernel in a seven-fold symmetric Ag$_{56}$ nanowheel

Zhi Wang[1], Hai-Feng Su[2], Mohamedally Kurmoo[3], Chen-Ho Tung[1], Di Sun [1] & Lan-Sun Zheng[2]

High-nuclearity silver clusters are appealing synthetic targets for their remarkable structures, but most are isolated serendipitously. We report here six giant silver-thiolate clusters mediated by solvents, which not only dictate the formation of an octahedral Ag$_6^{4+}$ kernel, but also influence the in situ-generated Mo-based anion templates. The typical sevenfold symmetric silver nanowheels show a hierarchical cluster-in-cluster structure that comprises an outermost Ag$_{56}$ shell and an inner Ag$_6^{4+}$ kernel in the centre with seven MoO$_4^{2-}$ anion templates around it. Electrospray ionization mass spectrometry analyses reveal the underlying rule for the formation of such unique silver nanowheels. This work establishes a solvent–intervention approach to construct high-nuclearity silver clusters in which both the formation of octahedral Ag$_6^{4+}$ kernel and in situ generation of various Mo-based anion templates can be simultaneously controlled.

[1] Key Laboratory of the Colloid and Interface Chemistry, Ministry of Education, and School of Chemistry and Chemical Engineering, Shandong University, 250100 Jinan, China. [2] State Key Laboratory for Physical Chemistry of Solid Surfaces and Department of Chemistry, College of Chemistry and Chemical Engineering, Xiamen University, 361005 Xiamen, China. [3] Institut de Chimie de Strasbourg, Université de Strasbourg, CNRS-UMR 7177, 4 Rue Blaise Pascal, 67008 Strasbourg Cedex, France. Correspondence and requests for materials should be addressed to D.S. (email: dsun@sdu.edu.cn)

High-nuclearity clusters of silver ions continue to fascinate chemists because of their intriguing geometrical characteristics such as high symmetry, large dimension and architectural beauty as well as some promising applications[1–3]. Recent advances in the silver cluster field show the subdivisions: type-I, the silver atoms exclusively aggregate to the cluster core and show an average oxidation state between 0 and 1[4–9]; type-II, the formation of metal core is not exclusively from the aggregation of monovalent silver but with the participation of small anions or/and organic ligands[10–16]. The representatives of type-I and -II clusters are $[Ag_{44}(p\text{-MBA})_{30}]^{4-}$ [17] and $[Ag_{490}S_{188}(StC_5H_{11})_{114}]$[18], respectively. These two kinds of silver clusters may be seen as molecular embryos of the bulk phases of silver metal and binary silver sulfides, respectively, and thus emerged as a hot frontier in chemistry and nanoscience research.

We have been working on the development of new strategies to access silver clusters with some level of control on the metal nuclearity and geometry for a given ligand type[19–24]. However, we must admit that almost all silver clusters were originally isolated serendipitously and the cases of deliberate modulation of nuclearity and geometry are very few, with most based on an anion template method. Among diverse anions, polyoxometalates (POMs) as a family of inorganic cluster have very rich structures and interesting properties[25–31]. Recent work showed the encapsulation of POMs into silver-thiolate/alkynyl clusters could effectively direct the nuclearity and geometry of the final products depending on the size and shape of the POMs [32–38]. Moreover, POMs usually isomerize or transform to other forms different from their original compositions and structures when enwrapped into the silver clusters[39], which should be controlled by the pH of the assembly environment and influenced by polarity of solvents[40]. On the other hand, solvents such as DMF (N,N-Dimethylformamide) contain aldehyde group that not only could endow electrons from O atom to form coordination bond with metal centre, but also has the ability to reduce Ag(I) to Ag(0)[41], which favours the formation of type-I silver clusters.

With these points in mind, herein, we show that it is quite effective to follow the solvent–intervention strategy to combine two types of silver clusters into one compound. They are sevenfold symmetry cluster-in-cluster silver nanowheels with an octahedral $Ag_6^{4+}$ kernel trapped in the centre. The family of SD/Ag7–SD/Ag10 (SD = SunDi) is the highest-nuclearity silver clusters possessing wheel-like topology and trapping the maximum $MoO_4^{2-}$ ions as template in one wheel. To justify the unique solvent effect in this assembly system, another two high-nuclearity silver-thiolate clusters (SD/Ag11 and SD/Ag12) are isolated with the $Ag_6^{4+}$ kernel template replaced by two novel POMs in the absence of DMF. This work pioneers a solvent–intervention approach to construct high-nuclearity silver nanowheels incorporating both an octahedral $Ag_6^{4+}$ kernel and a sevenfold symmetric silver shell.

## Results

**Structures of SD/Ag7–SD/Ag10.** Although $p\text{-TOS}^-$, $CF_3SO_3^-$ and $NO_3^-$ were employed as anions in the construction of SD/Ag7–SD/Ag10 from $CH_3OH/DMF$ (v:v = 4:1) as a binary solvent (Fig. 1), single-crystal X-ray diffraction (SCXRD) analyses showed that their sevenfold symmetric wheel-like topology was conserved, suggesting these anions have little effects on the cluster skeletons. Due to different molecules packing in the lattice (Supplementary Figure 1), they crystallize in different space groups, monoclinic $P2_1/c$, monoclinic $C2/c$, tetragonal $P4_2/ncm$ and monoclinic $I2/m$ space groups, respectively. Their formulae were determined as $[Ag_6@(MoO_4)_7@Ag_{56}(MoO_4)_2(^iPrS)_{28}(p\text{-TOS})_{14}(DMF)_4]$ (SD/Ag7), $[Ag_6@(MoO_4)_7@Ag_{56}(MoO_4)_2(^iPrS)_{28}(p\text{-TOS})_{14}\cdot3DMF]$ (SD/Ag8), $[Ag_6@(MoO_4)_7@Ag_{56}(MoO_4)_2(^iPrS)_{28}(CF_3SO_3)_{14}(DMF)_4]$ (SD/Ag9), $[Ag_6@(MoO_4)_7@Ag_{56}(MoO_4)_2(^iPrS)_{28}(NO_3)_{14}]$ (SD/Ag10). Because of the striking similarities of the molecules in SD/Ag7–SD/Ag10, only that of SD/Ag7 is described in detail here. Selected details of the data collection and structure refinements are listed in Supplementary Table 1.

The centrosymmetric silver wheel of SD/Ag7, as shown in Fig. 2a, like a motorcycle tyre, comprises of an $Ag_{56}$ shell and an octahedral $Ag_6^{4+}$ kernel in the centre with up to seven $MoO_4^{2-}$ as anion templates around it. Two additional $MoO_4^{2-}$ anions serve as hubcaps of the wheel. This wheel has a large dimension even without considering the organic ligands, measuring ~1.7 nm across its circular face (Fig. 2b) and 0.8 nm across the rectangular

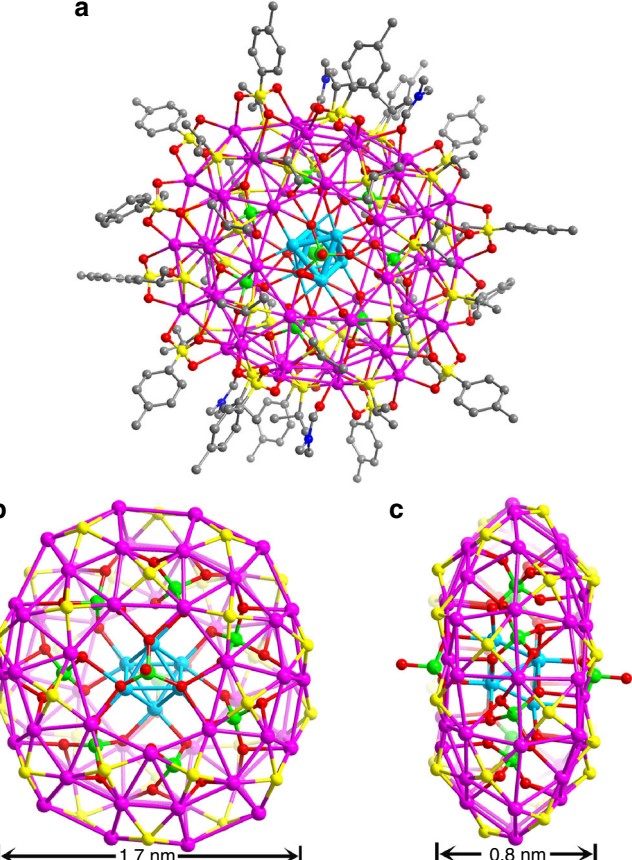

**Fig. 2** Single-crystal structure of **SD/Ag7**. **a** The total structure of **SD/Ag7** showed in ball-and-stick model. Ball-and-stick model of **SD/Ag7** showing the front (**b**) and side (**c**) views with all C atoms omitted for clarity. (Colour legend: purple, Ag on the shell; cyan, Ag in the kernel; yellow, S; red, O; green, Mo)

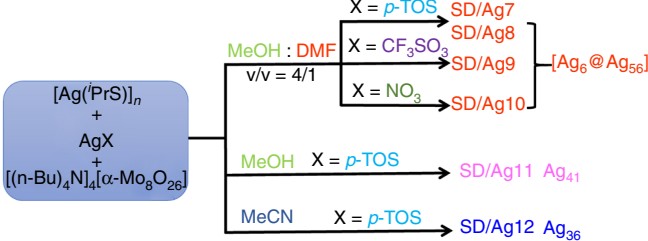

**Fig. 1** Schematic representation of the synthesis of clusters **SD/Ag7-SD/Ag12**. These clusters are formed by a solvent-controlled method

rim (Fig. 2c). It resembles the Preyssler-type polyoxometalate. In an asymmetric unit, a half of a wheel was identified and other half could be generated by inversion symmetry. The $Ag_{56}$ shell is covered by 28 $^iPrS^−$ and 14 $p$-TOS$^−$ ligands and consolidated by abundant Ag···Ag interactions (2.926–3.367 Å), forming 14 silver trigons and 42 silver tetragons fused together in an edge-sharing manner. In detail, on the circular face of the wheel, the silver trigons are isolated by silver tetragons and share their edges; whereas all silver tetragons are fused together to form the rectangular rim of the wheel (Supplementary Figure 2). All ligands exclusively capped on the tetragonal faces, where $\mu_4$ $^iPrS^−$, $\mu_4-\eta^2{:}\eta^1{:}\eta^1$ $p$-TOS$^−$ and $\mu_3-\eta^1{:}\eta^1{:}\eta^1$ $p$-TOS$^−$ were observed with the Ag–S and Ag–O distances ranging 2.374–2.552 and 2.400–2.770 Å, respectively.

Alternatively, we can also imagine this wheel as five concentric metallocycles of different diameters linked by $\mu_4$ $^iPrS^−$ ligands. The five metallocycles, nearly perpendicular to the pseudo sevenfold axis, contain one large $Ag_{14}$ ring at the equator ($\phi = 17.14$ Å; $d_{Ag···Ag} = 3.554–4.032$ Å) with two $Ag_7$ rings ($\phi = 7.68$ Å; $d_{Ag···Ag} = 3.366–3.797$ Å) and two small $Ag_{14}$ rings ($\phi = 13.66$ Å; $d_{Ag···Ag} = 2.926–3.331$ Å) lying above and below this ring (Fig. 3a). Two $Ag_7$ rings are not face-to-face stacking but rotated by an angle of 27.6° with respect to each other, whereas two small $Ag_{14}$ rings are almost face-to-face stacking (Supplementary Figure 3). Interestingly, we found that up to seven $MoO_4^{2−}$ anions were arranged along the equatorial $Ag_{14}$ ring inside (Fig. 3b). All seven $MoO_4^{2−}$ ions adopt the same $\mu_8-\eta^2{:}\eta^2{:}\eta^2{:}\eta^2$ bonding fashion to all $Ag_{14}$ rings as well as the inner octahedral

$Ag_6^{4+}$ kernel with an average Ag–O distance of 2.444 Å (Fig. 3c). This is the maximum number of $MoO_4^{2−}$ ions as template trapped in one cluster. The previous record was a 55-nuclearity silver-alkynyl cluster with six $MoO_4^{2−}$ ions inside[42]. Not only the number of Ag atoms in each ring, but also the number of $MoO_4^{2−}$ ions are 7 or multiples of 7, which ultimately leads to, albeit not rigorous, a sevenfold symmetry of the overall wheel. The overall geometry of such cluster shows typical anisotropy with rare high-order odd symmetry which should be dictated by initially prearranged seven $MoO_4^{2−}$ anions around $Ag_6^{4+}$ kernel, as justified by a series of $[Ag_6@(MoO_4)_7@Ag_n]$ ($n = 28–32$) intermediates observed in the electrospray ionization mass spectrometry (ESI-MS) of an early-stage reaction mixture during the synthesis of **SD/Ag7** (Supplementary Figure 4 and Supplementary Table 2). Generally, the orders of symmetry with a prime number equal to or higher than 5 are disfavoured[43], thus artificial molecular clusters with five- or sevenfold symmetry remain a rarity[44–46], although such symmetries usually exist in biomacromolecules such as HslV (heat shock locus V) and chaperone GroEL[47, 48]. The sevenfold symmetric characteristic is the first observed in silver clusters as exemplified by **SD/Ag7–SD/Ag10**.

Another structural feature is a centremost octahedral $Ag_6^{4+}$ kernel (Ag1, Ag2, Ag3 and their symmetry equivalents) which is formed purely by Ag···Ag interactions ranging from 2.659 to 2.852 Å (Fig. 3d). These short contacts between silver atoms are even shorter than those in bulk Ag metal (2.886 Å), indicating the significant argentophilicity[49–51]. The $Ag_6$ kernel is stabilized by a total of nine $MoO_4^{2−}$ anions through the Ag–O bonding (Fig. 3e) at the exposed [111] facets or edges of octahedron. This $Ag_6^{4+}$ kernel can be deemed as the smallest single-crystal octahedral silver nanocrystal[52] cut from the face-centred cubic (*fcc*) lattice of bulk silver but with a slight shrinkage due to the addition of two more electrons to the bonding orbitals[53]. Although several inorganic compounds, such as $Ag_6Ge_{10}P_{12}$[54], $Ag_6O_2$[55], $Ag_5SiO_4$[56] and $Ag_5GeO_4$[57], have been identified to contain such subvalent cluster unit, it is still rarely found in ligand-capped silver clusters[58, 59]. The formation of such partially reduced species should be caused by the reductive ability of DMF, which is widely used in the controlled synthesis of multiple-twin decahedral and icosahedral silver nanocrystals with special favourable [111] faces by reducing Ag$^+$ to Ag$^0$[41]. During this assembly process, DMF was partially oxidized to Me$_2$NCOOH, which was unambiguously verified by the $^{13}$C NMR (nuclear magnetic resonance) of HCl-digested reaction mother solution (Supplementary Figure 5). Based on above analyses and discussions, we found the reducing ability of DMF dictates the formation of the innermost $Ag_6^{4+}$ kernel, which then attracts seven $MoO_4^{2−}$ anions around it, thus forming the final sevenfold symmetric silver nanowheel. All these results clearly demonstrate that the reducibility of DMF is the key to such unique silver nanowheel.

Molecular wheels caught chemists' eyes as far back as 20-year ago since the giant POM wheels ($Mo_{154}$, $Mo_{176}$ and even larger $Mo_{248}$) were first characterized by Müller using X-ray diffraction[60]. Following their work, other giant wheel-like clusters, such as $Mn_{84}$[61], $Ni_{24}$[62] and $Pd_{84}$[63], were also successfully synthesized, suggesting such clusters were possible beyond Mo-based POMs. Among them, $Mo_{154}$ and $Pd_{84}$ are two limited sevenfold symmetric wheels. The wheel-like topology in silver clusters has not been reported up to now.

A careful examination of the packing of the wheels in the crystal lattice of **SD/Ag7** reveals that there are intermolecular C–H···O hydrogen bonding and van der Waals force between neighbours; the nearest Ag···Ag distance between neighbouring units is ~8.3 Å, with nearest neighbours oriented parallel to one another, that is, packing face to face (Supplementary Figure 1). However, the wheels packing showed an inclined rim-to-face

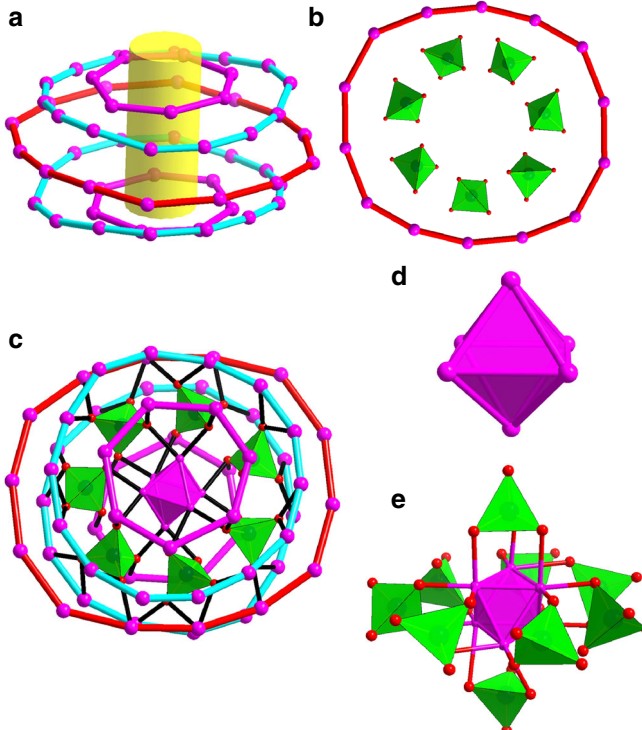

**Fig. 3** Disassembled skeletal structure of **SD/Ag7**. **a** Five concentric metallocycles: two $Ag_7$ (purple); two small $Ag_{14}$ (cyan) and one large $Ag_{14}$ (red). **b** Seven $MoO_4^{2−}$ anions arranged along the equatorial $Ag_{14}$ ring. **c** The bonding of $MoO_4^{2−}$ with $Ag_{14}$ rings and inner $Ag_6$ kernel with Ag–O bonding highlighted by black. **d** The polyhedral mode showing octahedral $Ag_6$ kernel. **e** The $Ag_6$ kernel protected by nine $MoO_4^{2−}$ anions with seven in the equator and two at the poles. (Colour legend: purple, Ag; red, O; green, Mo)

orientation for **SD/Ag8** and **SD/Ag9**, and a displaced face-to-face orientation for **SD/Ag10**.

**Structures of SD/Ag11 and SD/Ag12**. The comparative experiment to justify the reduction-induced formation of subvalent $Ag_6^{4+}$ kernel has resulted in the isolation of another two huge silver clusters **SD/Ag11** and **SD/Ag12** in the absence of DMF ([$Mo_7O_{24}$@$Ag_{41}$($^iPrS$)$_{19}$(p-TOS)$_{16}$(CH$_3$OH)$_4$·4CH$_3$OH] (**SD/Ag11**) and (n-Bu$_4$N)$_{1.5}$[$Mo_5O_{18}$@$Ag_{36}$($^iPrS$)$_{18}$(p-TOS)$_{13.5}$(CH$_3$CN)·1.5CH$_3$CN] (**SD/Ag12**)). In their centres, two novel POMs anions are trapped instead of the $Ag_6^{4+}$ kernel, suggesting the decisive role of DMF on the formation of subvalent $Ag_6^{4+}$ kernel.

When the solvent was changed to methanol, **SD/Ag11** crystallized in the orthorhombic space group *Pccn* and was isolated as a 41-nuclearity ellipsoidal silver shell loaded with a $C_{2v}$-$Mo_7O_{24}^{6-}$ in the centre as template (Fig. 4a). There are 19 $\mu_4$-$^iPrS^-$, 16 p-TOS$^-$ ($2 \times \mu_4$-$\eta^2$:$\eta^1$:$\eta^1$, $6 \times \mu_3$-$\eta^1$:$\eta^1$:$\eta^1$, $6 \times \mu_2$-$\eta^1$:$\eta^1$ and $2 \times \mu_4$-$\eta^2$:$\eta^2$), four CH$_3$OH ($2 \times \mu_2$ and $2 \times \mu_1$) molecules

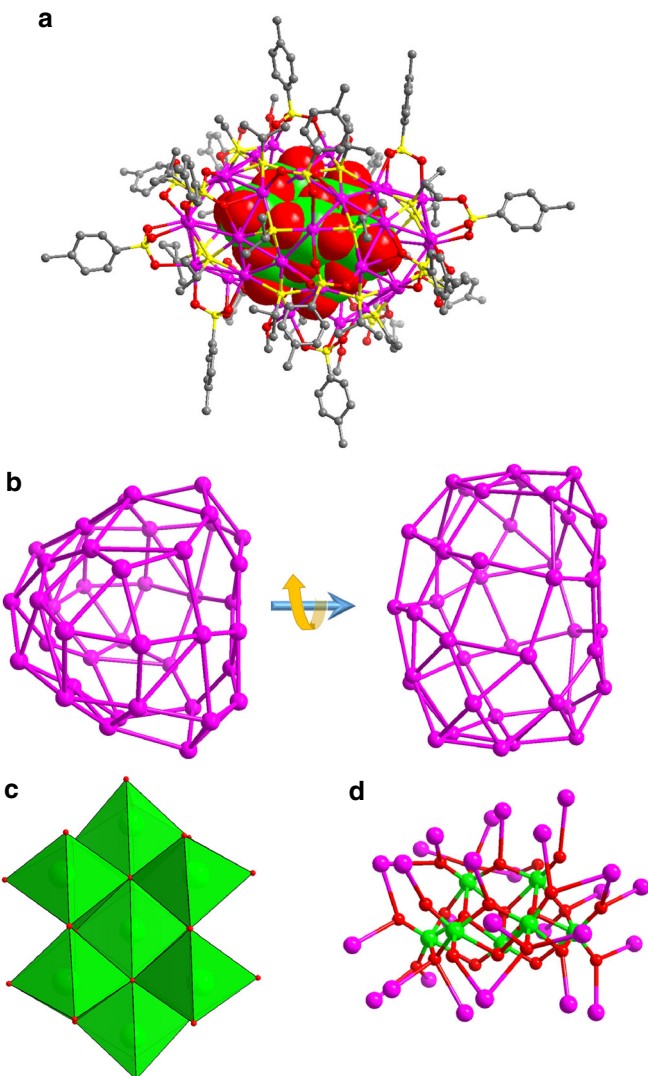

**Fig. 4** Single-crystal structure of **SD/Ag11**. **a** The total structure of **SD/Ag11** with $Mo_7O_{24}^{6-}$ shown in space filling mode. **b** The Ag$_{41}$ skeleton viewed along different directions. **c** Polyhedral representation of $Mo_7O_{24}^{6-}$. **d** The bonding between $Mo_7O_{24}^{6-}$ and Ag atoms at the shell. (Colour legend: purple, Ag; yellow, S; red, O; green, Mo)

coordinated on this irregular Ag$_{41}$ shell with the Ag–S and Ag–O distances of 2.388–2.532 and 2.338–2.592 Å. The abundant Ag···Ag interactions (2.886–3.374 Å) finally reinforce the overall shell (Fig. 4b). As shown in Fig. 4c, an $Mo_7O_{24}^{6-}$ ion was clearly resolved and the bond valence sum (BVS) calculations[64] indicate that all Mo atoms have an oxidation state of +6 (Mo1: 5.811; Mo2: 5.716; Mo3: 6.002 and Mo4: 6.030). All Mo atoms are coordinated to six O atoms, forming seven MoO$_6$ octahedra, which are fused to $Mo_7O_{24}^{6-}$ by exclusively edge-sharing fashion. The Mo–O bond lengths vary from 1.711 to 2.511 Å. The 41 silver atoms arranged around the surface of $Mo_7O_{24}^{6-}$ forming the core–shell structure. Among 41 silver atoms, 26 are coordinated to both terminal and bridge O atoms (Fig. 4d) with Ag–O distances in the range of 2.378–2.783 Å. The $Mo_7O_{24}^{6-}$ should be in situ generated from α-$Mo_8O_{26}^{4-}$.

When using CH$_3$CN as solvent, we isolated **SD/Ag12** as a 36-nuclearity silver cluster with an $Mo_5O_{18}^{6-}$ as core as revealed by SCXRD structural analysis (Fig. 5a). This anionic Ag$_{36}$ cluster can be shaped as a rugby-like skeleton (Fig. 5b) built from an equatorial Ag$_{18}$S$_{12}$ barrel adding two half-cuboctahedral Ag$_9$S$_3$ caps above and below, and they are joined together by argentophilic interactions and through Ag–S bonds (Fig. 5c). The Ag$_{36}$ shell is peripherally bridged by 18 $\mu_4$-$^iPrS^-$, 13.5 p-TOS$^-$ ($9 \times \mu_3$-$\eta^1$:$\eta^1$:$\eta^1$, $1 \times \mu_3$-$\eta^2$:$\eta^1$ and $3.5 \times \mu_2$-$\eta^1$:$\eta^1$) and one CH$_3$CN molecule. The Ag–S and Ag–O distances are located in the range of 2.430–2.637 and 2.293–2.569 Å, respectively. The Ag$_{36}$ skeleton consists of 8 trigons and 18 tetragons, which are edge-fused together to form the Ag$_{36}$ shell. The $\mu_4$-$^iPrS^-$ exclusively bonded to tetragons, whereas p-TOS$^-$ coordinated to both trigons on Ag$_9$S$_3$ caps and partial tetragons on Ag$_{18}$S$_{12}$ barrel. In **SD/Ag12**, Ag···Ag distances between 2.926 and 2.354 Å; are observed. The dimensions of the Ag$_{36}$ shell account for ~1.4 nm in length and 0.9 nm in width. The most fascinating feature of **SD/Ag12** is the inner $Mo_5O_{18}^{6-}$ core, which is built from two octahedral MoO$_6$, two square-pyramidal MoO$_5$ and one tetrahedral MoO$_4$ units (Fig. 5d). The 36 silver atoms aggregated around the surface of $Mo_5O_{18}^{6-}$ forming the core–shell structure. Among 36 silver atoms, 25 are coordinated to both terminal and bridge O atoms (Fig. 5e). Such $Mo_5O_{18}^{6-}$ may be transient and can only be stabilized in the void of silver cluster through the rich Ag–O bonding.

From the above structural analyses, we found that (i) the same POM precursor can transform to different POMs in different solvents (MeOH vs MeCN); (ii) Mo-based anions with rich geometries and compositions templated diverse silver clusters; (iii) DMF plays an important role in the reductive formation of subvalent silver kernel; (iv) multiple simple and small anion templates (MoO$_4^{2-}$) can also induce the formation of large silver clusters through special arrangement of the anions. The multiple roles of solvents promise rational access to more complex and diverse silver clusters with special geometries or symmetries.

**Electrospray ionization mass spectrometry**. To detect the stability of the cluster, the ESI-MS of **SD/Ag8**, **SD/Ag9** and **SD/Ag11** dissolved in acetonitrile were performed in positive-ion mode. As depicted in Fig. 6a, b, **SD/Ag8** displays five identifiable species (**2a–2e**), whereas **SD/Ag9** shows six species (**3a–3f**) with the charge state of +4 in the range of m/z = 1800–2400. After carefully analyzing these peaks in the ESI-MS of **SD/Ag8** and **SD/Ag9**, we did not find any parent [$Ag_6$@(MoO$_4$)$_7$@$Ag_{56}$] species, which means neither **SD/Ag8** nor **SD/Ag9** are stable in acetonitrile, however, fragments with the [$Ag_6$@(MoO$_4$)$_7$] core were unambiguously detected. The dominating species (**2b** and **3e**), centred at m/z = 1952.2731 and 2010.2228, can be assigned to [$Ag_6$@(MoO$_4$)$_7$@$Ag_{35}$($^iPrS$)$_{11}$(p-TOS)$_6$Cl

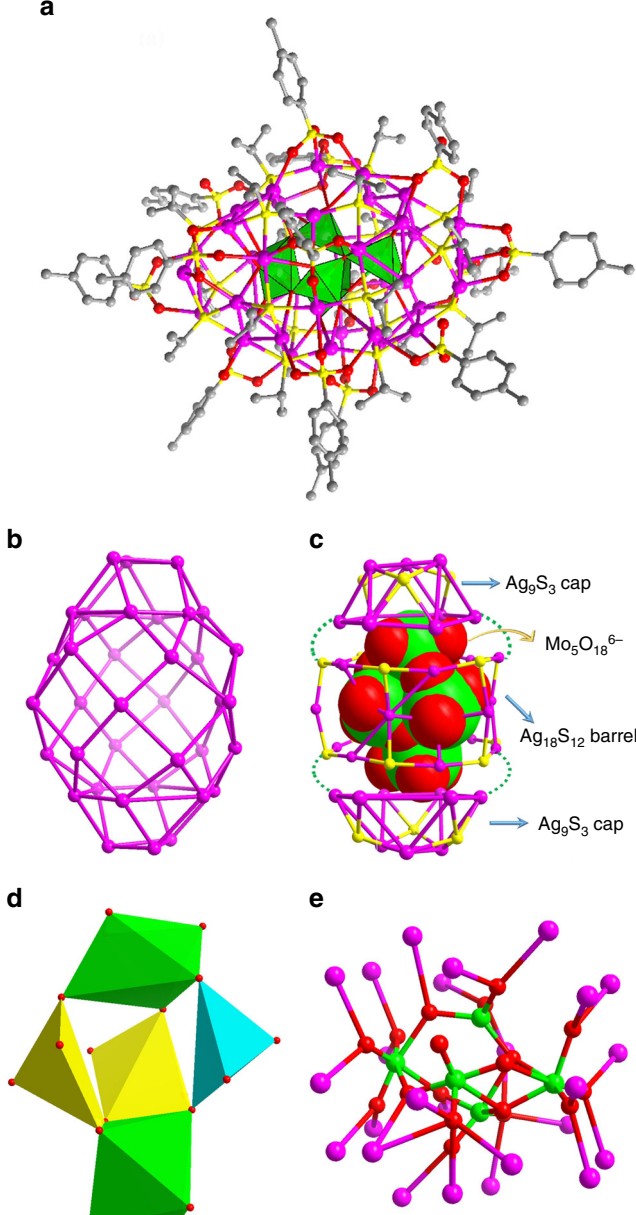

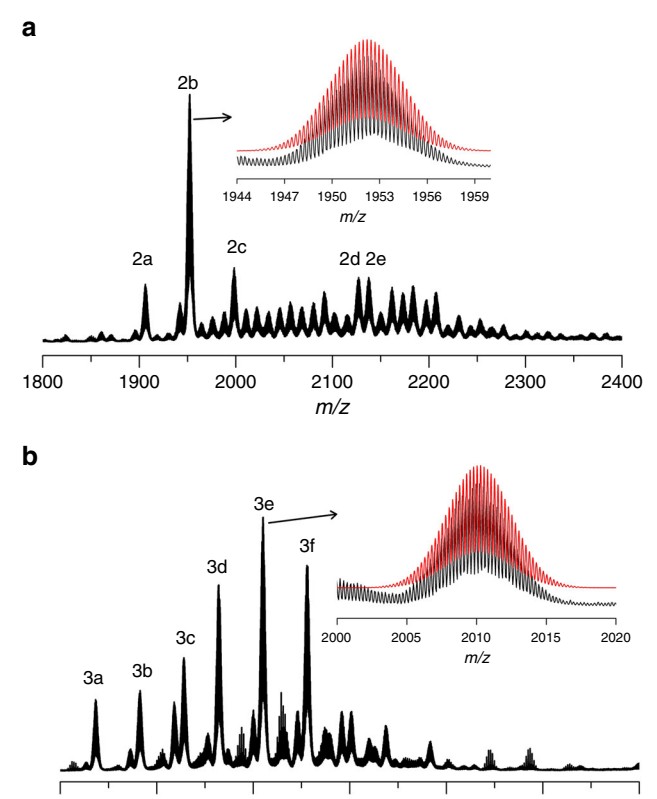

**Fig. 6** The solution behaviours of **SD/Ag8** and **SD/Ag9**. Positive-mode ESI-MS of **SD/Ag8** (**a**) and **SD/Ag9** (**b**) dissolved in acetonitrile. Insets: Comparison of the experimental (black spectrum) and simulated (superimposed red spectrum) isotopic envelopes for **2b** and **3e**

**Fig. 5** Single-crystal structure of **SD/Ag12**. **a** The total structure of **SD/Ag12** with $Mo_5O_{18}^{6-}$ shown in polyhedral mode. **b** The $Ag_{36}$ skeleton comprised of $Ag_3$ trigons and $Ag_4$ tetragons. **c** Schematic showing one $Mo_5O_{18}^{6-}$ enwrapped by $Ag_{18}$ barrel, then closed by two half-cuboctahedral $Ag_9S_3$ caps. **d** Polyhedral representation of $Mo_5O_{18}^{6-}$ with two octahedral $MoO_6$, two square-pyramidal $MoO_5$ and one tetrahedral $MoO_4$ units shown in green, yellow and cyan, respectively. **e** The bonding between $Mo_5O_{18}^{6-}$ and Ag atoms at the shell. (Colour legend: purple, Ag; yellow, S; red, O; green, Mo)

$(OH)_3(CH_3CN)_4(H_2O)_9]^{4+}$ (Calc. $m/z = 1952.2102$) and $[Ag_6@(MoO_4)_7@Ag_{36}(^iPrS)_{13}(CF_3SO_3)_9(H_2O)_4]^{4+}$ (Calc. $m/z = 2010.2902$), respectively. The other species in the ESI-MS of **SD/Ag8** and **SD/Ag9** were identified based on the experimental and theoretical isotope distributions and presented in the Supplementary Figures 6, 7 and Supplementary Tables 3, 4. All species **2a–2e** and **3a–3f** have cores of $[Ag_6@(MoO_4)_7@Ag_n]$ ($n_{2a-2e} = 32 – 36$ and $n_{3a-3f} = 32 – 37$), which indicates that the innermost $[Ag_6@(MoO_4)_7]$ core in **SD/Ag8** and **SD/Ag9** is stable in solution, despite the absence of the complete $Ag_{56}$ parent shell.

Considering the charge state of the assigned species, the subvalent nature of $Ag_6^{4+}$ was further justified by the ESI-MS results. With respect to outer ligand shell, the $\Delta m/z$ between **3a** and **3b** is 45.48, which is similar to the $\Delta m/z$ (45.98) between **3b** and **3c**, and the mass difference corresponds to one $Ag^iPrS$, indicating the coordination–dissociation equilibrium with losing or adding $Ag^iPrS$ one by one in solution. Based on the above results, the presence of seven $MoO_4^{2-}$ anions enwrapping an $Ag_6$ kernel should be precursor species, which is then closed by the outer $Ag_{56}$ wheel at a second step in the growth process. The overall hierarchical cluster-in-cluster growth process was thus established for $Ag_{62}$ cluster family, which was further justified by analyzing the reaction mixture by ESI-MS before crystallization (Supplementary Figure 4 and Supplementary Table 2).

As shown in Fig. 7, the species found in ESI-MS of **SD/Ag11** dissolved in acetonitrile could be divided into two groups based on their charge states, +3 (**5a–5s**) and +2 (**5t–5z**), respectively, in the range of $m/z = 2700–5100$. The most abundant peak in the triply-charged species (see the inset of Fig. 7) located at $m/z = 2910.2894$ (**5 g**), which can be assigned to $[Mo_7O_{24}@Ag_{41}(^iPrS)_{18}(p\text{-}TOS)_9Cl_5(CH_3CN)_4(H_2O)]^{3+}$ (Calc. $m/z = 2910.2880$) and the most abundant peak in doubly-charged species centred at $m/z = 4592.8767$ (**5w**), which can be attributed to $[Mo_7O_{24}@Ag_{41}(^iPrS)_{10}(p\text{-}TOS)_{15}Cl_6(OH)_2(CH_3CN)_3(H_2O)]^{2+}$ (Calc. $m/z = 4592.8338$). The other 24 species were also identified and are shown in the Supplementary Figure 8 and Supplementary Table 5. Compared to the crystallographic result of **SD/Ag11**, we discovered that all species have the identical $[Mo_7O_{24}@Ag_{41}$

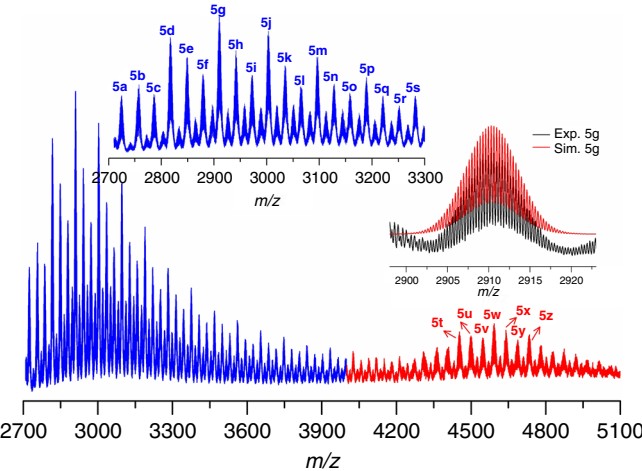

**Fig. 7** Positive-ion ESI-MS of **SD/Ag11** dissolved in acetonitrile. Insets: The enlarged MS of triply-charged species (blue line) and comparison of the experimental (black spectrum) and simulated (superimposed red spectrum) isotopic envelopes for **5g**

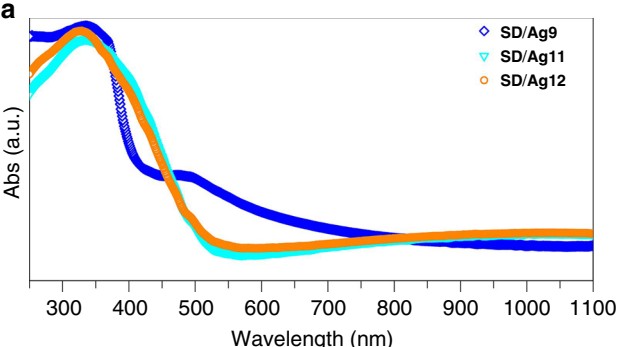

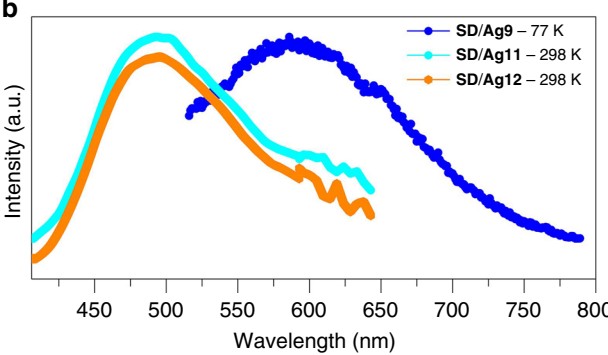

**Fig. 8** The spectral properties of **SD/Ag9**, **SD/Ag11** and **SD/Ag12**. The solid-state UV–Vis (**a**) and emission spectra (**b**) of the clusters **SD/Ag9**, **SD/Ag11** and **SD/Ag12**

skeleton, which indicates the core of **SD/Ag11** is intact in acetonitrile.

**UV–Vis absorption spectra and luminescence properties**. The solid-state optical diffuse reflectance spectra of **SD/Ag9**, **SD/Ag11** and **SD/Ag12** were investigated at room temperature. As shown in Fig. 8a, compound **SD/Ag9** exhibits one main peak centred at 338 nm and weak shoulder peak at 483 nm, which should be assigned to the $n \rightarrow \pi^*$ transition of $^i$PrS$^-$ and ligand-to-metal charge transfer (LMCT) transition, respectively. However, compounds **SD/Ag11** and **SD/Ag12** show only one peak at 335 and 328 nm, respectively, both belongs to the $n \rightarrow \pi^*$ transition of $^i$PrS$^-$.

Furthermore, the photoluminescence properties of **SD/Ag9**, **SD/Ag11** and **SD/Ag12** were also studied in the solid state. As shown in Fig. 8b, compound **SD/Ag9** is emission silent at room temperature, however, it emits yellow light at 77 K with a maximum at 587 nm ($\lambda_{ex} = 400$ nm), which is probably due to a ligand-to-metal charge transfer (LMCT, charge transfer from the S 3p to Ag 5s orbital) transition disturbed by Ag···Ag interactions[65]. This temperature-sensitive emission behaviours should be related to the variations of molecule rigidity and argentophilicity under different temperatures and have been previously observed in other silver-thiolate clusters such as [(CO₃)@Ag₂₀(SBu$^t$)₁₀(DMF)₆(NO₃)₈][66] and [S@Ag₅₆S₁₂(SBu$^t$)₂₀](CH₃COO)₁₀[67]. Compounds **SD/Ag11** and **SD/Ag12** emit green light at room temperature with a similar maximum emission peak at *ca.* 493 nm upon the excitation at 330 nm, which should be attributed to the LMCT transition, or mixed with Ag···Ag interactions.

## Discussion

In summary, we discovered a family of sevenfold symmetric silver nanoclusters featuring the anisotropic wheel-like geometry encapsulating an octahedral nanofragment of metallic silver inside. The formation of such nanoclusters was realized by solvent modulation, which not only dictates the formation of octahedral Ag₆$^{4+}$ kernel, but also influence the in-situ-generated Mo-based templates. Different anions, p-TOS$^-$, CF₃SO₃$^-$ and NO₃$^-$, were involved in four wheel-like silver-thiolate clusters but without breaking the basic wheel-like backbones. The overall Ag₆₂

nanocluster features a hierarchical cluster-in-cluster structure comprising an octahedral Ag₆$^{4+}$ kernel in the centre with up to seven MoO₄$^{2-}$ as templates around it, supporting the outermost Ag₅₆ shell. The formation of an inner octahedral Ag₆$^{4+}$ kernel is closely dependent on the DMF as reducing agent. The unique sevenfold odd symmetry should be dictated by the prearranged seven MoO₄$^{2-}$ templates around the Ag₆$^{4+}$ kernel. In the absence of DMF, two different high-nuclear silver-thiolate clusters were identified without the Ag₆$^{4+}$ kernel trapped in the inner space but instead two POM templates were found and in situ generated from the same starting POM precursor. The present study launches a more rational approach to construct high-nuclearity silver clusters in which both the formation of octahedral Ag₆$^{4+}$ kernel and in situ generation of various Mo-based anion templates is controlled by the solvents.

## Methods

**Synthesis of the (Ag$^i$PrS)ₙ**. The precursor of (Ag$^i$PrS)ₙ was prepared by the following reported procedure[20]. The solution of AgNO₃ (30 mmol, 5.1 g) in 75 mL acetonitrile was mixed with 100 mL ethanol containing $^i$PrSH (30 mmol, 2.8 mL) and 5 mL Et₃N under stirring for 3 h in the dark at room temperature, then the yellow powder of (AgS$^i$Pr)ₙ was isolated by filtration and washed with 10 mL ethanol and 20 mL ether, then dried in the ambient environment (yield: 97%).

**Synthesis of SD/Ag7 and SD/Ag8**. p-TOSAg (0.1 mmol, 27.9 mg) and (Ag$^i$PrS)ₙ (0.05 mmol, 9.2 mg) together with [(n-C₄H₉N)₄[α-Mo₈O₂₆] (0.0002 mmol, 4.2 mg) were dissolved in a mixed solvent of methanol: N,N′-dimethylformamide (5 mL, v/v = 4/1). The reaction mixture was sealed and heated at 65 °C for 2000 min, and then cooled to room temperature for 800 min. Then, the brown solution was filtered and the filtrate left to evaporate slowly for 2 weeks in the dark at room temperature. Brown block crystals of **SD/Ag7** and diamond crystals of **SD/Ag8** were crystallized in the yields of 10% and 13%, respectively.

Elemental analyses calc. (found) for **SD/Ag7**: C₁₉₄H₃₂₂Ag₆₂Mo₉N₄O₈₂S₄₂: C, 18.03 (18.12); H, 2.51 (2.59); N, 0.43 (0.39)%. Selected IR peaks (cm$^{-1}$): 2910 (w), 1441 (w), 1374 (w), 1248 (w), 1144 (m), 1115 (m), 1003 (m), 781 (m), 677 (s), 595 (s), 555 (s).

Elemental analyses calc. (found) for **SD/Ag8**: $C_{191}H_{315}Ag_{62}Mo_9N_3O_{81}S_{42}$: C, 17.86 (17.79); H, 2.47 (2.51); N, 0.33 (0.31)%. Selected IR peaks (cm$^{-1}$): 2934 (w), 1621 (w), 1486 (w), 1451 (w), 1373 (w), 1245 (w), 1153 (s), 1117 (s), 1004 (s), 784 (s), 677 (s), 600 (w), 564 (s).

**Synthesis of SD/Ag9**. The synthetic condition was similar to that described for **SD/Ag7** and **SD/Ag8**, except that the *p*-TOSAg was substituted by CF$_3$SO$_3$Ag (0.1 mmol, 25.6 mg), brown block crystals of **SD/Ag9** were crystallized in a yield of 35% after 3 days.

Elemental analyses calc. (found) for **SD/Ag9**: $C_{110}H_{224}Ag_{62}F_{42}Mo_9N_4O_{82}S_{42}$: C, 17.86 (17.90); H, 2.47 (2.55); N, 0.32 (0.29)%. Selected IR peaks (cm$^{-1}$): 2952 (w), 1643 (w), 1448 (w), 1373 (w), 1224 (m), 1167 (m), 1011 (m), 784 (m), 635 (s), 511 (m).

**Synthesis of SD/Ag10**. The synthetic condition was similar to that described for **SD/Ag7** and **SD/Ag8**, except that the *p*-TOSAg was replaced by AgNO$_3$ (0.1 mmol, 17 mg), brown crystals of compound **SD/Ag10** were crystallized in a yield of 20% after 3 weeks.

Elemental analyses calc. (found) for **SD/Ag10**: $C_{84}H_{196}Ag_{62}Mo_9N_{14}O_{78}S_{28}$: C, 9.09 (9.13); H, 1.78 (1.69); N, 1.77 (1.79)%. Selected IR peaks (cm$^{-1}$): 2948 (w), 1650 (w), 1526 (w), 1451 (w), 1387 (m), 1267 (m), 1146 (m), 1047 (m), 776 (s), 600 (m).

**Synthesis of SD/Ag11**. The synthetic condition was similar to that described for **SD/Ag7** and **SD/Ag8** but using MeOH (5 mL) instead, yellow rod crystals of **SD/Ag11** were isolated by filtration, washed with EtOH and dried in air (yield: 35%).

Elemental analyses calc. (found) for **SD/Ag11**: $C_{177}H_{277}Ag_{41}Mo_7O_{80}S_{35}$: C, 21.47 (21.51); H, 2.82 (2.79) %. Selected IR peaks (cm$^{-1}$): 2927 (w), 1600 (w), 1500 (w), 1444 (w), 1380 (w), 1249 (w), 1110 (s), 1004 (s), 869 (m), 841 (m), 805 (m), 677 (s), 613 (m), 550 (s).

**Synthesis of SD/Ag12**. The synthetic condition was similar to that described for **SD/Ag7** and **SD/Ag8** but using MeCN (5 mL) instead, yellow block crystals of compound **SD/Ag12** were crystallized in a yield of 20% after 10 days.

Elemental analyses calc. (found) for **SD/Ag12**: $C_{177.5}H_{283}Ag_{36}Mo_5N_4O_{58.5}S_{31.5}$: C, 24.28 (24.19); H, 3.25 (3.28); N, 0.64 (0.61) %. Selected IR peaks (cm$^{-1}$): 2934 (w), 1600 (w), 1495 (w), 1448 (w), 1382 (w), 1241 (w), 1117 (m), 1004 (s), 876 (w), 812 (m), 677 (s), 600 (m), 556 (s).

**Data availability**. The X-ray crystallographic coordinates for structures reported in this article have been deposited at the Cambridge Crystallographic Data Centre, under deposition number CCDC: 1815301-1815304, 1815357 and 1815305 for SD/Ag7-SD/Ag12. These data can be obtained free of charge from the Cambridge Crystallographic Data Centre via www.ccdc.cam.ac.uk/data_request/cif.

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

## Acknowledgements

This work was financially supported by the NSFC (Grant No. 21571115), the Natural Science Foundation of Shandong Province (Nos. JQ201803 and ZR2017MB061), Young Scholars Program of Shandong University (2015WLJH24), and the Fundamental Research Funds of Shandong University (104.205.2.5 and 2015JC045). M.K. is funded by the CNRS-France.

## Author contributions

Original idea was conceived by D.S., experiments and data analyses were performed by Z. W. and D.S., ESI-MS data were collected by H.-F.S., structure characterization was performed by Z.W. and D.S.; manuscript was drafted by D.S., Z.W., M.K., C.-H.T. and L.-S.Z. All authors have given approval to the manuscript.

## Additional information

**Competing interests:** The authors declare no competing interests.

