## [Peer Review File · Nature Communications]

Reviewers' comments:

Reviewer #1 (Remarks to the Author):

Reconsider after major revision.

The authors presented several novel silver polyoxometalate clusters, which deepens the understanding of such bulk materials or nanoparticles. Although the seven-fold symmetry around Ag₆ unit is fascinating, the authors fail to provide adequate evidences to reveal the origin of such high-order symmetry. The reductive ability of DMF is another interesting hypothesis, yet no strong proof can support the exact role during the reaction. Without showing why "the deliberate modulation of nuclearity and geometry" can be achieved under such conditions through strong experimental support, I think the manuscript suits a specialist journal better.

There is also a small concern for the mass spectrum of species 3a, 3b, 3c. The $\Delta m/z$ between 3a and 3b is 45.48, while $\Delta m/z$ between 3b and 3c is 45.98. The authors claim that the differences are the same (AgPrS). However, all other data from mass spectrometry have an error of around or less than 0.05 unit, except this one has an error of 0.5 unit. The assignment could be wrong.

Reviewer #2 (Remarks to the Author):

The submission by Sun et al. reports the solvothermal syntheses and structural determination by single-crystal X-ray diffraction of 6 novel silver thiolate clusters. The UV-Vis absorption and photoluminescence of 3 of the 6 clusters were also included, but the results appear to be not unusual.

All 6 clusters are structurally attractive, and the four featuring a central core of Ag₆ encapsulated by a cage of Ag₅₆ are just fascinating. The core and the cage are linked by 7 templating [MoO₄] anions that were generated in situ. As such, the beautiful cluster structure can be viewed as a giant Ag wheel with an approximate 7-fold rotational symmetry - a rarity in cluster chemistry. The formation of different Mo-based templating anions and the assembly of the Ag clusters obtained appear to be dictated by the solvent used in the solvothermal preparation. Mass spectroscopic evidence indicated that the cluster core together with the Mo-based templates maintain its structural integrity in solution.

The most significant contribution of this work to the field of metal cluster chemistry, that of Ag clusters in particular, is the demonstration of how solvents may play a critical role in modulating the assembly of the cluster species eventually obtained. Though different Mo-based templates were also observed, it is likely the formation of the Mo-based anion templates and the corresponding Ag clusters is mutually dependent. Profound solvent effects in cluster assembly and in chemical transformations are not new and in fact, a commonly accepted phenomenon. However, the results reported in this work are still rather striking. Nevertheless, exactly how these different solvents influence the reaction pathway and by doing so, dictating the reaction outcome was not discussed at all in the manuscript. At a couple of places, the reducing power of DMF was alluded to, but no details were offered. Could any of the suspected by-product(s) be detected to shed some light on the mechanism possibly responsible for the solvent effects? This would be the most significant scientific insight possibly provided by this work as only by doing so could the so-called controlled assembly of the clusters be a possibility. There is the potential, but relevant evidence needs to be offered unambiguously.

In terms of the presentation of the work, the manuscript would be enhanced significantly with some

language improvement.

Reviewer #3 (Remarks to the Author):

Four seven-fold symmetric wheel-like silver-thiolate clusters (denoted as SD/Ag7-SD/Ag10) showed a hierarchical 'cluster-in-cluster' structure that comprises an octahedral Ag₆₄₊ cluster in the center with up to seven MoO₄²⁻ as anion templates around it and with the outmost Ag₅₆ shell, and two high-nuclear silver-thiolate clusters (SD/Ag11 and SD/Ag12) were also isolated. Although the contribution by Sun et al. is technically well done, I do not recommend acceptance for publication in Nature Communications for the following reasons:

1) The chemistry described is too specialized as to be of interest for the broad readership of the journal.

2) The single crystal data needs to be as good as possible. However, the wR₂ values of the complexes 1-5 are quite high (wR₂ = 0.2505 for complex 1, wR₂ = 0.2654 for complex 2, wR₂ = 0.3327 for complex 3, wR₂ = 0.3822 for complex 4, and wR₂ = 0.2729 for complex 5).

Reviewer 1:

Reconsider after major revision.

(1) The authors presented several novel silver polyoxometalate clusters, which deepens the understanding of such bulk materials or nanoparticles. Although the seven-fold symmetry around Ag₆ unit is fascinating, the authors fail to provide adequate evidences to reveal the origin of such high-order symmetry. The reductive ability of DMF is another interesting hypothesis, yet no strong proof can support the exact role during the reaction. Without showing why "the deliberate modulation of nuclearity and geometry" can be achieved under such conditions through strong experimental support, I think the manuscript suits a specialist journal better.

Response: Thanks for these insightful suggestions. We are glad that the reviewer finds the present structures of clusters interesting and scientifically significant from the symmetry point of view. This section mainly contains three questions about this work. We totally agree with the reviewer that the manuscript could be improved by additional evidences that could unambiguously support i) the origin of 7-fold symmetry, ii) reductive ability of DMF and iii) the deliberate modulation of nuclearity and geometry of silver clusters.

(1) In order to access the origin of such high-order symmetry, we re-performed the electrospray ionization mass spectrometry (ESI-MS) measurement for the reaction mother liquor to extract the related information about the assembly process. Using this powerful characterization technique, we can detect a series of intermediates before the formation of the final 7-fold symmetry Ag₆₂ cluster. These results are very positive as we expected. We observed a series of [Ag₆@(MoO₄)₇@Ag_n] (n = 28 - 32) intermediates in the ESI-MS of reaction mother liquor (see below figure or Fig. S4 in SI), which suggested the initial trapping of seven MoO₄²⁻ by some Ag_n fragments happened before the final formation of [Ag₆@(MoO₄)₇@Ag₅₆]. The annular arrangement of seven MoO₄²⁻ anion should be the result of stabilizing the bare highly active Ag₆⁴⁺ kernel through surface Ag-O interactions. Why seven and not six or eight assuming to go with the six corners or eight faces of the octahedral Ag₆

remains a mystery. We speculate it may be to do with space matching. Thus, we believed that the rare high-order odd symmetry should be caused by initially prearranged seven MoO_4^{2-} around the Ag_6^{4+} kernel. The related discussions were added into the structure description section as ‘*The overall geometry of such cluster shows typical anisotropy with rare high-order odd symmetry which should be dictated by initially prearranged seven MoO_4^{2-} anions around Ag_6^{4+} kernel, as justified by the a series of $[\text{Ag}_6@(\text{MoO}_4)_7@\text{Ag}_n]$ ($n = 28 - 32$) intermediates observed in the electrospray ionization mass spectrometry (ESI-MS) of the reaction mother liquor (Fig. S4).’*

Peaks	Species	Exp. m/z	Sim. m/z
A	$[\text{Ag}_6@(\text{MoO}_4)_7@\text{Ag}_{28}(\text{}^i\text{PrS})_{13}(p\text{-TOS})\text{Cl}(\text{H}_2\text{O})(\text{CH}_3\text{OH})_2]^{3+}$	2017.7840	2017.7851
B	$[\text{Ag}_6@(\text{MoO}_4)_7@\text{Ag}_{29}(\text{}^i\text{PrS})_{13}(p\text{-TOS})_2\text{Cl}(\text{H}_2\text{O})(\text{CH}_3\text{OH})_2]^{3+}$	2110.4260	2110.4240
C	$[\text{Ag}_6@(\text{MoO}_4)_7@\text{Ag}_{30}(\text{}^i\text{PrS})_{13}(p\text{-TOS})_3\text{Cl}(\text{H}_2\text{O})(\text{CH}_3\text{OH})_2]^{3+}$	2203.7278	2203.7295
D	$[\text{Ag}_6@(\text{MoO}_4)_7@\text{Ag}_{31}(\text{}^i\text{PrS})_{13}(p\text{-TOS})_4\text{Cl}(\text{H}_2\text{O})(\text{CH}_3\text{OH})_2]^{3+}$	2296.7001	2296.7017
E	$[\text{Ag}_6@(\text{MoO}_4)_7@\text{Ag}_{32}(\text{}^i\text{PrS})_{13}(p\text{-TOS})_5\text{Cl}(\text{H}_2\text{O})(\text{CH}_3\text{OH})_2]^{3+}$	2389.6739	2389.6708
F	$[\text{Ag}_6@(\text{MoO}_4)_7@\text{Ag}_{29}(\text{}^i\text{PrS})_{16}\text{Cl}(\text{H}_2\text{O})(\text{C}_3\text{H}_7\text{NO})]^{2+}$	3111.6822	3111.6649
G	$[\text{Ag}_6@(\text{MoO}_4)_7@\text{Ag}_{30}(\text{}^i\text{PrS})_{16}(p\text{-TOS})\text{Cl}(\text{H}_2\text{O})(\text{C}_3\text{H}_7\text{NO})]^{2+}$	3251.6416	3251.6232

(2) Reductive ability of DMF is well known in the controlled synthesis of multiple-twin decahedral and icosahedral silver nanocrystals with special favourable [111] faces (*Nat Mater* 11, 131-137 (2012); *Nano Lett* 2, 903-905

(2002); *J Cryst Growth* 289, 376-380 (2006); *Cryst Growth Des* 10, 5238-5243 (2010); *Cryst Growth Des* 10, 296-301 (2010); *Cryst Growth Des* 9, 4700-4705 (2009); *Langmuir* 18, 5981-5983 (2002)). The reductive ability mainly from its aldehyde group, similar to the famous ‘silver mirror’ reaction. The reaction occurred as ‘ $\text{HCONMe}_2 + 2\text{Ag}^+ + \text{H}_2\text{O} \rightarrow 2\text{Ag}^0 + \text{Me}_2\text{NCOOH} + 2\text{H}^+$ ’ (*Pure Appl Chem* 72, 83-90 (2000)). In spite of so famous behaviour, we still followed the comment of this reviewer to add solid proof to support above hypothesis using ^{13}C NMR. After solvothermal reaction, the peak corresponding to Me_2NCOOH was found at 161.96 (see below figure or Fig. S5). The related discussions were also added into revised manuscript as ‘The formation of such partially reduced species should be caused by the reductive ability of DMF, which is widely used in the controlled synthesis of multiple-twin decahedral and icosahedral silver nanocrystals with special favourable [111] facets by reducing Ag^+ to Ag^0 .’⁴¹ During this assembly process, DMF was partially oxidized to Me_2NCOOH , which was unambiguously verified by the ^{13}C NMR (nuclear magnetic resonance) of HCl digested reaction mother solution (Fig. S5).’.

(3) As compared structures of six novel silver clusters, the deliberate modulation of nuclearity and geometry of silver clusters was obvious and are summarized into the last paragraph of structure description section as *'From the above structural analyses of the six silver-thiolate clusters, we found that (i) the same POM precursor can transform to different POMs in different solvents (MeOH vs MeCN); (ii) Mo-based anions have rich geometries and compositions to act as anion template in the construction of diverse silver clusters; (iii) DMF plays an important role in the reductive formation of subvalent silver nanoclusters; (iv) multiple simple and small anion templates (MoO_4^{2-}) can also induce the formation of large silver clusters through special arrangement of the anions. The multiple roles of solvents promise new and rational access to more complex and diverse silver clusters with special geometries or symmetries.'* The more deeply modulations for silver cluster assembly is definitely an important follow-up research topic to be pursued with significant efforts.

Based on the above additional experiments and explanations, we hope the revised manuscript suits *Nat. Comm.* better with wide broad interests.

(2) There is also a small concern for the mass spectrum of species 3a, 3b, 3c. The $\Delta m/z$ between 3a and 3b is 45.48, while $\Delta m/z$ between 3b and 3c is 45.98. The authors claim that the differences are the same (AgPrS). However, all other data from mass spectrometry have an error of around or less than 0.05 unit, except this one has an error of 0.5 unit. The assignment could be wrong.

Response: Thanks for the insightful comment about the ESI-MS analysis. We are sorry for the confusion caused by less-detailed interpretation of MS data in the original submission. We carefully reconsidered above assignment for the m/z differences between adjacent species. As shown in Figure S6, the perfect matching between black and red lines indicated the correct assignment for each species. Please note, we showed the m/z values in manuscript uniformly using the molecular weight of 100% abundance isotope instead of the average molecular weight. So the experimental $\Delta m/z$ between 3a and 3b is 45.4804 (cal. 45.4830), while $\Delta m/z$ between 3b and 3c is 45.9837 (cal. 45.9829). These two

delta_m/z values are exactly matched with 100% and 97.49% abundance isotopes of [AgPrS]⁴⁺ (see below figure). So such errors are caused by the shift of 100% abundance isotopes when adding one more AgPrS into one species. We also clarified the m/z readout value in SI material as ‘The reported m/z values represent monoisotopic mass of the most abundant peak (100%) within the isotope pattern.’ and other detailed ESI-MS measurement conditions also were added into SI material.

Reviewer 2:

(1) The submission by Sun et al. reports the solvothermal syntheses and structural determination by single-crystal X-ray diffraction of 6 novel silver thiolate clusters. The UV-Vis absorption and photoluminescence of 3 of the 6 clusters were also included, but the results appear to be not unusual.

Response: Thanks for these insightful suggestions. We are glad that the reviewer finds the present structures of clusters interesting and scientifically significant. Including the interesting structure and ESI-MS result, we also tried to find some properties for these cluster compounds, so some properties such as optical absorption and luminescence were incorporated into the manuscript. If suitable, we can move this section into SI section, which will not shade the shining points of this work.

(2) All 6 clusters are structurally attractive, and the four featuring a central core of Ag₆ encapsulated by a cage of Ag₅₆ are just fascinating. The core and the cage are linked by 7 templating [MoO₄] anions that were generated in situ. As such, the beautiful cluster structure can be viewed as a giant Ag wheel with an approximate 7-fold rotational symmetry - a rarity in cluster chemistry. The formation of different Mo-based templating anions and the assembly of the Ag clusters obtained appear to be dictated by the solvent used in the solvothermal preparation. Mass spectroscopic evidence indicated that the cluster core together with the Mo-based templates maintain its structural integrity in solution.

Response: We are pleased and excited by the reviewer's positive acknowledgement on the novelty and significance of our study. The assembly of more interesting silver clusters in a controllable way is definitely a good research topic of our follow-up study, which requires significant efforts on the systematic investigation.

(3) The most significant contribution of this work to the field of metal cluster chemistry, that of Ag clusters in particular, is the demonstration of how solvents may play a critical role in modulating the assembly of the cluster species eventually obtained. Though different Mo-based templates were also observed, it is likely the

formation of the Mo-based anion templates and the corresponding Ag clusters is mutually dependent. Profound solvent effects in cluster assembly and in chemical transformations are not new and in fact, a commonly accepted phenomenon. However, the results reported in this work are still rather striking. Nevertheless, exactly how these different solvents influence the reaction pathway and by doing so, dictating the reaction outcome was not discussed at all in the manuscript. At a couple of places, the reducing power of DMF was alluded to, but no details were offered. Could any of the suspected by-product(s) be detected to shed some light on the mechanism possible responsible for the solvent effects? This would be the most significant scientific insight possibly provided by this work as only by doing so could the so-called controlled assembly of the clusters be a possibility. There is the potential, but relevant evidence needs to be offered unambiguously.

Response: We are thankful for the reviewer's constructive suggestions/comments. This concern about the proposed DMF-reduction route to the formation of Ag₆₂ cluster also was referred by reviewer 1. We should acknowledge this reviewer and the first one's constructive comments/suggestions, which have encouraged us to deepen our mechanistic understandings of the DMF reduction reaction in our assembly system. Thus, we especially addressed this case with some additional experiments as well as references. We are sorry that we did not articulate well in the previous version about the important solvent effect as shown in scheme 1. Reductive ability of DMF is well known in the controlled synthesis of multiple-twin decahedral and icosahedral silver nanocrystals with special favourable [111] faces (*Nat Mater* 11, 131-137 (2012); *Nano Lett* 2, 903-905 (2002); *J Cryst Growth* 289, 376-380 (2006); *Cryst Growth Des* 10, 5238-5243 (2010); *Cryst Growth Des* 10, 296-301 (2010); *Cryst Growth Des* 9, 4700-4705 (2009); *Langmuir* 18, 5981-5983 (2002)). The reductive ability mainly from its aldehyde group, similar to the famous 'silver mirror' reaction. The reaction occurred as 'HCONMe₂ + 2Ag⁺ + H₂O → 2Ag⁰ + Me₂NCOOH + 2H⁺' (*Pure Appl Chem* 72, 83-90 (2000)). In spite of so famous behaviour, we still followed the comment of this reviewer to add solid proof to support above hypothesis

using ^{13}C NMR. After solvothermal reaction, the peak corresponding Me_2NCOOH was found at 161.96 (see below figure). The related discussions were also added into revised manuscript as *'The formation of such partially reduced species should be caused by the reductive ability of DMF, which is widely used in the controlled synthesis of multiple-twin decahedral and icosahedral silver nanocrystals with special favourable [111] facets by reducing Ag^+ to $\text{Ag}^{0.41}$. During this assembly process, DMF was partially oxidized to Me_2NCOOH , which was unambiguously identified by the ^{13}C NMR (nuclear magnetic resonance) of HCl digested reaction mother solution (Fig. S5).'*

(4) In terms of the presentation of the work, the manuscript would be enhanced significantly with some language improvement.

Response: Thanks for your constructive suggestion. We revised several typos and awkward phrasing by shortening several long sentences for easily reading. The further English polishing was also performed with the help from the native English speaker.

Reviewer 3:

Four seven-fold symmetric wheel-like silver-thiolate clusters (denoted as SD/Ag7-SD/Ag10) showed a hierarchical ‘cluster-in-cluster’ structure that comprises an octahedral Ag₆⁴⁺ cluster in the center with up to seven MoO₄²⁻ as anion templates around it and with the outmost Ag₅₆ shell, and two high-nuclear silver-thiolate clusters (SD/Ag11 and SD/Ag12) were also isolated. Although the contribution by Sun et al. is technically well done, I do not recommend acceptance for publication in Nature Communications for the following reasons:

(1) The chemistry described is too specialized as to be of interest for the broad readership of the journal.

Response: Thanks for this reviewer’s concerns about the broad interest of this work. We are sorry that we did not articulate well in the previous version about the broad interest of this work. In the revised manuscript, we comprehensively revised our manuscript suitable to several research fields such as crystal engineering, coordination chemistry, cluster chemistry, early transition metal compound, POMs, interface chemistry, nanomaterial, anion templating synthesis, supramolecular assembly, ESI-MS of coordination compound, crystallography and so on. For example, i) the molecular Ag₆⁴⁺ kernel is the smallest single-crystal octahedral silver nanocrystal cut from the face-centred cubic (*fcc*) lattice of bulk silver. Such nanocrystal was interfacially stabilized by multiple molybdate anions. This involves coordination chemistry, POMs, interface chemistry, nanomaterial, and anion templating synthesis; ii) solvent-modulated structures belongs to crystal engineering, supramolecular assembly and crystallography; iii) the final luminescent properties also relate to material chemistry; iv) The ESI-MS used in such large clusters is not recognized often enough in the community, and definitely has the ability to provide insights into growth mechanisms for such large clusters (Also see our recent related work: *Proc Natl Acad Sci USA* 2017, 114, 12132; *J. Am. Chem. Soc.* 2016, 138, 1328; *J. Am. Chem. Soc.* 2017, 139, 14033; *J. Am. Chem. Soc.* 2018, 140, 1600). In all, we

hope that this revised version will attract broad interest from wide research fields and will allow him/her to come to a positive recommendation. Thank you!

(2) The single crystal data needs to be as good as possible. However, the wR2 values of the complexes 1-5 are quite high (wR2 = 0.2505 for complex 1, wR2 = 0.2654 for complex 2, wR2 = 0.3327 for complex 3, wR2 = 0.3822 for complex 4, and wR2 = 0.2729 for complex 5).

Response: Thank you very much for your remarks regarding crystallography. The structural determinations of these large clusters, especially for those with high symmetry, have been problematic for several reasons. The absorption correction is difficult for crystals containing high electron atoms (Ag, Mo) and low electron atoms (C, O, N, H) in the cell. The weak high-angle Bragg diffractions limit the resolution of the structures and the brittle nature of the crystals due to the rapid loss of co-crystallized solvents once leaving mother liquor, through every precautions are taken to reduce this chemical effect. For these reasons the quality of the crystal structures is not too high. In our case, especially for four Ag₆₂ clusters, not only inner Ag₆ kernel but also the MoO₄²⁻ ions show multiple disorders (2-3 different orientations), which further deteriorated the crystal quality and the final refinements.

Anyway, we had tried our best to grow large enough crystals of high quality and had also attempted to collect better data sets by use of new Rigaku Oxford Diffraction XtaLAB Synergy diffractometer equipped with a HyPix-6000HE area detector, using a Mo K_α radiation from PhotonJet micro-focus X-ray source. We are happy to see the significant improvement in the new datasets. The wR2 values of the complexes 1-5 are considerably decreased and all R1 values for complexes 1-5 are now below 10%. We have used the new data to replace all the old ones in main text as well as SI material and hope this reviewer will be satisfied with these new datasets. The compared refinement results were summarized in table below.

	R ₁ and wR ₂	SD/Ag7	SD/Ag8	SD/Ag9	SD/Ag10	SD/Ag11
Original	Final R	R ₁ = 0.0938	R ₁ = 0.1006	R ₁ = 0.1185	R ₁ = 0.1181	R ₁ = 0.1161

Submission	indexes [$I \geq 2\sigma(I)$]	$wR_2 = 0.2365$	$wR_2 = 0.2431$	$wR_2 = 0.3014$	$wR_2 = 0.3399$	$wR_2 = 0.2563$
	Final R indexes [all data]	$R_1 = 0.1148$ $wR_2 = 0.2505$	$R_1 = 0.1369$ $wR_2 = 0.2654$	$R_1 = 0.1666$ $wR_2 = 0.3327$	$R_1 = 0.1630$ $wR_2 = 0.3822$	$R_1 = 0.1484$ $wR_2 = 0.2729$
Revised Submission	Final R indexes [$I \geq 2\sigma(I)$]	$R_1 = 0.0854$ $wR_2 = 0.2148$	$R_1 = 0.0873$ $wR_2 = 0.1969$	$R_1 = 0.0818$ $wR_2 = 0.2111$	$R_1 = 0.0892$ $wR_2 = 0.2417$	$R_1 = 0.0450$ $wR_2 = 0.0963$
	Final R indexes [all data]	$R_1 = 0.0998$ $wR_2 = 0.2248$	$R_1 = 0.1134$ $wR_2 = 0.2120$	$R_1 = 0.0982$ $wR_2 = 0.2274$	$R_1 = 0.1059$ $wR_2 = 0.2585$	$R_1 = 0.0588$ $wR_2 = 0.1072$

REVIEWERS' COMMENTS:

Reviewer #1 (Remarks to the Author):

The authors did a wonderful job to addressing the major concerns.

However, it might be better to point out that the ESI-MS in Figure S4 is obtained from an early-stage reaction mixture. Otherwise, it could be just another Figure S6 or S7, where all peaks are from the fragmentation or decomposition.

Although evidence implies that the reducing ability of DMF is the key to the seven-fold symmetry, there is no straightforward statement to confirm or deny or even discuss this. The authors could add a few sentences to discuss it.

The manuscript can be accepted with minor revision. No further review is needed.

Reviewer #2 (Remarks to the Author):

I have now carefully examined the present revised manuscript. The questions and concerns raised in my original review have all been addressed in a satisfactory manner, in particular toward the mechanistic details of the role played by DMF. I need to point out that the same concern was also raised by Reviewer 1. With the additional experiments, solid evidence in support of the oxidation of DMF was established.

I have also read through the responses by the authors to the comments by Reviewers 1 and 3. In my opinion, the authors have made genuine efforts to address those concerns/questions. The only remaining and tough-to-answer question is why the 7-fold symmetry was generated. The authors speculated that it could be due to the steric or the so-called space requirement around the central Ag₆ kernel; it certainly could be one of the reasons, and I am not sure how one may prove it.

Considering the above-mentioned improvement made with the revisions, I feel comfortable in recommending the acceptance of this revised submission with some further language polishing. Of course the editorial office will take good care of that.

Reviewer #3 (Remarks to the Author):

The points raised in the previous round of review have been satisfactorily addressed. The wR₂ values of the complexes 1-5 are considerably decreased and all R₁ values for complexes 1-5 are now below 10%. These new datasets are good. The revised manuscript is suitable for publication.

Reviewer #1 (Remarks to the Author):

The authors did a wonderful job to addressing the major concerns.

Response: Thank you very much. We appreciate your recommendation for publication of our work in Nature Communications.

However, it might be better to point out that the ESI-MS in Figure S4 is obtained from an early-stage reaction mixture. Otherwise, it could be just another Figure S6 or S7, where all peaks are from the fragmentation or decomposition.

Response: Thanks for your insightful suggestion. We have revised the caption of Figure S4 to “Supplementary Figure 4: The simulated and experimental isotopic distributions of species found in ESI-MS of an early-stage reaction mixture during the synthesis of SD/Ag7.”

Although evidence implies that the reducing ability of DMF is the key to the seven-fold symmetry, there is no straightforward statement to confirm or deny or even discuss this. The authors could add a few sentences to discuss it.

Response: Thanks for your constructive suggestion. We have added two sentences to discuss this issue as “Based on above analyses and discussions, we found the reducing ability of DMF dictates the formation of the innermost Ag_6^{4+} kernel, which then attracts seven MoO_4^{2-} anions around it, thus forming the final 7-fold symmetric silver nanowheel. All these results clearly demonstrate that the reducibility of DMF is the key to such unique silver nanowheel.”

The manuscript can be accepted with minor revision. No further review is needed.

Response: We appreciate the reviewer’s inputs and have revised our manuscript according to these comments. We hope our revised manuscript is now suitable for publication in Nature Communications.

Reviewer #2 (Remarks to the Author):

I have now carefully examined the present revised manuscript. The questions and concerns raised in my original review have all been addressed in a satisfactory manner, in particular toward the mechanistic details of the role played by DMF. I need to point out that the same concern was also raised by Reviewer 1. With the additional experiments, solid evident in support of the oxidation of DMF was established.

I have also read through the responses by the authors to the comments by Reviewers 1 and 3. In my opinion, the authors have made genuine efforts to address those concerns/questions. The only remaining and tough-to-answer question is why the 7-fold symmetry was generated. The authors speculated that it could be due to the steric or the so-called space requirement around the central Ag₆ kernel; it certainly could be one of the reasons, and I am not sure how one may prove it.

Considering the above-mentioned improvement made with the revisions, I feel comfortable in recommending the acceptance of this revised submission with some further language polishing. Of course the editorial office will take good care of that.

Response: We really appreciate such encouraging remarks by the reviewer. We carefully rechecked the overall manuscript and SI, and made some corrections on the language. Moreover, the language was further polished by one of our co-authors, Prof. Mohamed. We hope our revised manuscript is now suitable for publication in *Nature Communications*.

Reviewer #3 (Remarks to the Author):

The points raised in the previous round of review have been satisfactorily addressed. The wR2 values of the complexes 1-5 are considerably decreased and all R1 values for complexes 1-5 are now below 10%. These new datasets are good. The revised manuscript is suitable for publication.

Response: Thank you very much. We appreciate your recommendation for publication of our work in Nature Communications.